# Inference-Friendly Models With MixAttention

## Abstract

The size of the key-value (KV) cache plays a critical role in determining both the maximum context length and the number of concurrent requests supported during inference in modern language models. The KV cache size grows proportionally with the number of attention heads and the tokens processed, leading to increased memory consumption and slower inference for long inputs. In this work, we explore the use of MixAttention, a model architecture modification closely related to a blog published by Character.AI (Character.AI, 2024). MixAttention combines sliding window attention, where only a small subset of recent tokens is stored in the KV cache, with KV cache sharing across layers. Our experiments demonstrate that MixAttention significantly reduces memory usage and improves inference speed without sacrificing model performance in both short and long-context tasks. We also explore various configurations of this architecture, identifying those that maintain quality across evaluation metrics while optimizing resource efficiency.

## 1 Introduction

Transformer-based language models are getting increasing popular in consumer usage as well as industrial workloads. A general trend seen so far has been that bigger models are better at tasks than smaller models, but that comes at the cost of increased inference cost and slower speed (Hoffmann et al., 2022; Sardana et al., 2024). Further, the memory consumption and latency during inference for causal attention transformer models like Llama (Touvron et al., 2023; Dubey et al., 2024), GPT (Radford et al., 2019), and Gemini (Team et al., 2023) increases linearly with the input length. This causes problems for use cases such as Retrieval Augmented Generation (RAG) (Lewis et al., 2020), where the input to the models can become very long (Leng et al., 2024).

An important component of the Transformer architecture whose memory footprint grows with model size and input length is its KV cache. When generating the next output token, the transformer model processes all the tokens in its context through the attention mechanism. For causal attention models, the internal representation of the previous tokens in the context is unaffected by the newer tokens, and hence it can be cached. This is stored in the KV cache, and its size increase with context length (since it caches information for each token seen so far) and with the size of the model (since there is a separate KV cache for each KV head in the model). Larger KV cache not only means more memory consumption by the model, but it also slows down inference because for long inputs, LLM inference can be dominated by the I/O cost of moving the KV cache from HBM to the GPU's shared memory. Thus, it has become imperative to reduce the size of the KV cache for faster and cost-effective inference with modern LLMs.

Several methods have been proposed for reducing the KV cache size including sparse attention methods (Beltagy et al., 2020), reducing the number of KV heads (Ainslie et al., 2023; Shazeer, 2019), KV quantization (Hooper et al., 2024), inference-time cache sparsification through token eviction (Zhang et al., 2024), or even replacing some of the attention layers with State Space Machine (SSM) layers (Lieber et al., 2024). Most of these methods are compatible with others, for example using GQA with Sliding Window Attention (Jiang et al., 2023), using GQA with quantization (Hooper et al., 2024; Lin et al., 2024), or interleaving SSM layers with Sliding Window Attention layers (Lieber et al., 2024; Ren et al., 2024). In this paper, we explore such a combination proposed by Character.AI where they combine Sliding Window Attention with KV cache sharing across layers (Character.AI, 2024). We train and evaluate several variants of this architecture, and find that differ-

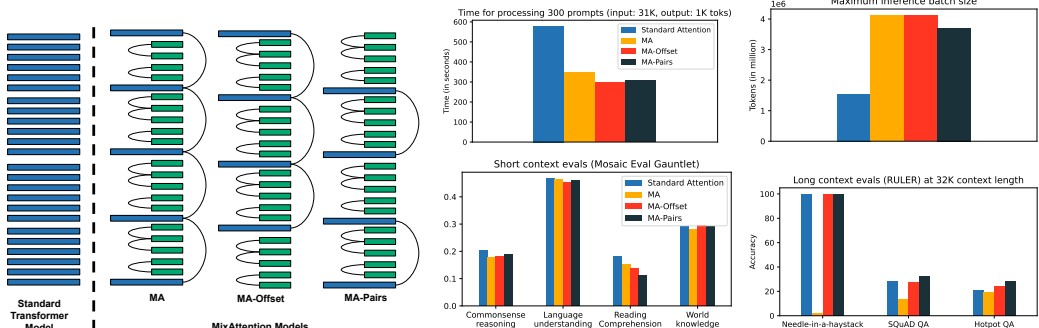

**Figure 1:** (Left) Variants of MixAttention architecture - green bars represent sliding window attention and the curved lines connecting bars represent KV cache sharing. (Right, top row) We see that MixAttention models are faster and use less memory during inference at 32K context length. (Right, bottom row) MixAttention models maintain quality - they match the standard attention model on most evals. The models are all Mixture of Experts with 2B active and 5B total parameters.

ent ways of combining the two ideas result in very different model abilities. In particular, we find some configurations that match the standard transformer model in most short and long context evals, while being faster and more memory efficient during inference.

### 1.1 CONTRIBUTIONS

We find that KV cache sharing between layers and adding sliding window layers speeds up inference and reduces inference memory usage while maintaining model quality, although some eval metrics show some degradation (Figure 1). In addition, our ablation experiments show the following:

- Having a few standard attention layers is crucial for the model's long context abilities. In particular, having the standard KV cache computed in the deeper layers is more important for long context abilities than the standard KV cache of the first few layers.
- KV cache of standard attention layers can be shared between non-consecutive layers without any observed degradation in long context abilities.
- Increasing the KV-cache sharing between sliding window layers too much also hurts the long-context abilities.

## 2 RELATED WORK

Reducing the KV cache size has been an area of active research, with many different approaches. In this section we talk about some of them.

**Linear Attention and SSM Models.** Transformer models (Vaswani, 2017) differ from traditional Recurrent Neural Networks (RNNs) (Sherstinsky, 2020) and modern State Space Models (Gu et al., 2022; Ren et al., 2024) in that Transformer models have an internal representation (the KV cache) that grows linearly with the length of the input. This allows RNNs and SSMs to be faster and more memory efficient during inference. However, it has been seen that while such models are competitive with Transformer models on certain tasks, Transformer models still outperform equally-sized pure RNN or pure SSM models on other tasks, especially some long context tasks (Waleffe et al., 2024). Thus, hybrid architectures which interleave attention layers and SSM layers have been proposed, that show that such hybrid architectures exhibit good long context abilities (Lieber et al., 2024; Ren et al., 2024). Other works have linearized the attention mechanism by replacing the softmax operation with kernelized similarity computation, showing both speed and memory improvements for inference (Katharopoulos et al., 2020).

**KV Quantization.** KV quantization works by reducing the precision of the cached key-value (KV) pairs which reduces the overall storage requirements and improves the data movement efficiency during inference (Lin et al., 2024). Hooper et al. (2024) combined several novel methods for

quantizing the KV cache to achieve significant improvements in long-context language model performance, achieving context lengths up to 10 million tokens while maintaining performance metrics close to those of unquantized models.

**KV Eviction.** KV eviction dynamically remove less relevant or older tokens from the KV cache during inference to reduce its size. This approach ensures that only the most pertinent tokens are retained, helping alleviate memory bottlenecks in long-context tasks. Zhang et al. (2024) proposed Heavy-Hitter Oracle ($H_2O$), an efficient generative inference approach that selectively evicts tokens from the cache based on their relevance, significantly improving the performance of models operating on large contexts. Chen et al. (2024) introduced NaCl which generalizes $H_2O$ and adds random token eviction to retain long context performance while evicting tokens.

**KV Head Reduction.** Architectures like Multi-Query Attention (MQA) (Shazeer, 2019) and Grouped Query Attention (GQA) (Ainslie et al., 2023) show that the number of KV heads in the attention layer can be decreased without significantly impacting model performance. Multi-Query Attention (MQA) simplifies the standard Multi-Head Attention mechanism (Vaswani, 2017) by sharing the key and value projections across all attention heads in a layer while retaining independent queries. This approach drastically reduces the size of the KV cache, as fewer unique key-value pairs are stored during inference. However, when serving models on multiple GPUs using Tensor Parallelism (TP) (Shoeybi et al., 2019), the single key and value cache must be replicated across the tensor parallel ranks, thus essentially losing a significant fraction of the memory savings gained from using MQA. Hence, Grouped Query Attention (GQA) extends the same idea of query sharing but instead of having one set of keys and values for all the queries, this architecture partitions queries into multiple sets and shares keys and values within each set, where the number of sets (and hence the number of keys and values) often matches the TP rank.

**Sparse and Local Attention.** Sparse and Local Attention mechanisms have been extensively explored as a means to improve the efficiency of Transformer models by reducing the quadratic complexity of traditional global attention. Since these methods focus on attending to only a subset of tokens, they reduce the computational and memory costs during both training and inference. One of the most prominent methods in this category is Longformer (Beltagy et al., 2020), which introduces several variants of local attention mechanism including Sliding Window Attention. Sliding Window Attention and its variants restrict the attention of each token to a fixed window of neighboring tokens, rather than all tokens in the sequence, drastically reducing the number of key-value pairs that need to be stored and processed during inference. This method has been shown to work well but often fails on tasks with long-context dependencies due to the fundamental lack of global attention. Sparse Attention mechanisms further optimize the attention computation by introducing sparsity patterns, where only certain key-value pairs are attended to based on predefined criteria (Beltagy et al., 2020). Notably, GPT-3 used interleaving global and local attention layers in its architecture (Brown, 2020).

**KV Sharing.** KV Sharing (Brandon et al., 2024; Wu and Tu, 2024) is a key technique used to reduce the memory footprint of Transformer models during inference by allowing multiple layers to reuse the same key-value (KV) instead of having separate KV pairs for each layer. Brandon et al. (2024) demonstrate that cross-layer attention, where KV caches are shared across different layers, leads to substantial memory savings without degrading accuracy.

## 3 MixAttention

Standard transformer models use global attention in each layer. To create inference-friendly model architectures, we use a combination of sliding window attention layers, standard attention, and KV cache reuse layers. Below is a brief discussion on each component:

**Sliding Window Attention Layers (Beltagy et al., 2020):** In Sliding Window Attention (or Local Attention) with window size $s$, the query only pays attention to the last $s$ keys instead of all the keys preceding it. This means that during inference, the KV cache size needs to only store the KV tensors for the past $s$ tokens instead of storing the KV tensors for all the preceding tokens. In our experiments, we set a window size of $s = 1024$ tokens.

**Standard Attention Layers:** We found that even though Standard Attention Layers lead to bigger KV caches and slower attention computation compared to Sliding Window Attention, having a few Standard Attention Layers is crucial for the model's long context abilities.

**KV cache reuse (Brandon et al., 2024; Wu and Tu, 2024):** This refers to a layer in the transformer network reusing the KV cache computed by a earlier layer. Hence, if every $l$ layers share KV tensors, then the size of KV cache is reduced by factor of $1/l$.

We experimented with different combinations of the components above to ablate the effects of each of them (Figure 2). We found that not only do each of the above components play important roles in long context abilities and inference speed and memory consumption, but also their relative positions and counts have significant effects on those metrics.

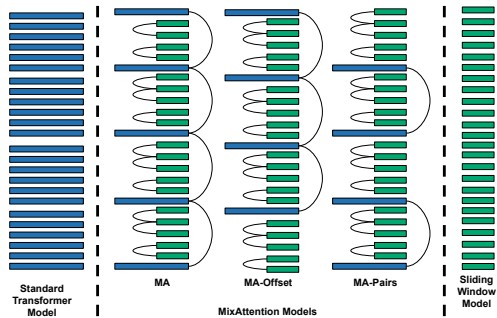

**Figure 2:** MixAttention: (Left) A standard transformer model where all layers are standard attention layers. (Middle) Inference-friendly models with Mix-Attention. Green bars represent sliding window attention and the lines connecting bars represent KV cache sharing. (Right) A model where all layers are sliding window attention.

The models we trained are 24-layer Mixture of Experts (MoE) models with 1.64B active and 5.21B total parameters. We used RoPE positional embeddings (Su et al., 2024), and increased the RoPE base theta as we increased the context length during training. We used Grouped Query Attention (Ainslie et al., 2023) with 12 attention heads and 3 KV heads.

## 4 EXPERIMENTS

### 4.1 TRAINING

We used LLM Foundry (Mosaic, 2023b) to train MixAttention models. Similar to prior work on training long context models [5, 6], we followed a multi-stage training procedure to impart long context abilities to the models.

1. We pretrained the models with a RoPE theta of 0.5M on 101B tokens, where each sequence sequence has been truncated to 4k token length.

2. To increase the context length, we then trained the model on 9B tokens on a mix of natural language and code data, where the sequences have been truncated to 32k tokens. We increased the RoPE theta to 8M for this stage. When training at 32k context length (*i.e.*, this step and the next step), we trained only the attention weights and froze the rest of the network. We found that this delivered better results than full network training.

3. Finally, we trained the model on a 32K-length, synthetic, long-context QA dataset [5, 8].
   - To create the dataset, we took natural language documents and chunked them into 1k-token chunks. Each chunk was then fed to a pretrained instruction model and the model was prompted to generate a question-answer pair based on the chunk. Then, we concatenated chunks from different documents together to serve as the "long context." At the end of this long context, the question-answer pairs for each of the chunks were added. The loss gradients were computed only on the answer parts of these sequences.
   - This phase of training was conducted on 500M tokens (this number includes the tokens from the context, questions, and answers). The RoPE theta was kept at 8M for this stage.

### 4.2 EVALUATION

The models were evaluated on the Mosaic Evaluation Gauntlet v 0.3.0 (Mosaic, 2023a) to measure model quality across various metrics including reading comprehension, commonsense reasoning, world knowledge, symbolic problem solving, and language understanding. To evaluate the models'

long context abilities, we used RULER (Hsieh et al., 2024) at a context length of 32000 tokens. RULER is a composite benchmark consisting of 13 individual evals of the following types:

- Needle-in-a-haystack (NIAH): These types of evals hide a single or multiple keys and values in a long text, and the model is evaluated on its ability to retrieve the correct value(s) from the long context for a given key(s).

- Variable Tracking (VT): This eval provides the model with a long context containing variable assignment statements, and the model is tasked to figure out which variables have a particular value by the end of all the variable assignments.

- Common and Frequent Word Extraction (CWE and FWE): These tasks ask the model to extract the most common or frequent words from the text.

- Question Answering (QA): Given a long context, the model is asked a question from somewhere in the context and the model is evaluated on whether it can correctly answer that question.

We used SGLang (Zheng et al., 2023) to deploy our models on 1 NVIDIA H100 GPU to run RULER and get inference speed and memory consumption metrics.

## 5 RESULTS

### 5.1 POSITION AND COUNT OF STANDARD ATTENTION KV CACHES

To measure the effect of the position and count of the standard attention KV caches, we tried four variants (Figure 3). All the configurations are variants of the configuration proposed in Character.AI's blog (Character.AI, 2024).

*MA*: This variant has a single standard attention KV cache, which is the KV cache of the first layer. All the other standard attention layers share this KV cache.

*MA-EndSlide*: This variant is the same as MA, but the last layer is a sliding window attention layer. This was done to measure how much having standard attention in the last layer affects long-context abilities.

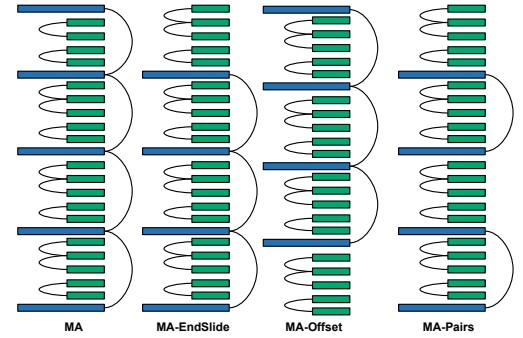

**Figure 3:** KV Cache position and counts: To measure the effect of the position and count of the standard attention KV caches on MixAttention's long context abilities, we train and evaluate the 4 models shown above.

*MA-Offset*: This variant is similar to MA, but the first standard attention layer is offset to a later layer to allow the model to process the local context for a few layers before the standard attention layer is used to look at longer contexts.

*MA-Pairs*: This variant computes two standard attention KV caches (at the first and thirteenth layer), which are then shared with another standard attention layer each.

We compared these models to a transformer model with Standard Attention and a transformer model with Sliding Window Attention in all layers.

While the loss curves in Stages 1 and 2 of training were close for all the models, we found that in Stage 3 (training on long context QA dataset), there was a clear bifurcation in the loss curves (Figure 4, top). In particular, we see that configurations MA and MA-EndSlide show much worse loss than the others. These results are consistent with the long context RULER evals, where we found that MA and MA-EndSlide performed much worse than others (Figure 4, bottom). Their performance was similar to the performance of the network with only sliding window attention in all layers. We think the loss in Stage 3 correlates well with RULER evals because unlike Stages 1 and 2, which were next-word prediction tasks where local context was sufficient to predict the next word most of the time, in Stage 3 the model needed to retrieve the correct information from potentially long-distance context to answer the questions. As we see from the RULER evals, MA-Offset and MA-Pairs have better long-context abilities than MA and MA-EndSlide across all the categories.

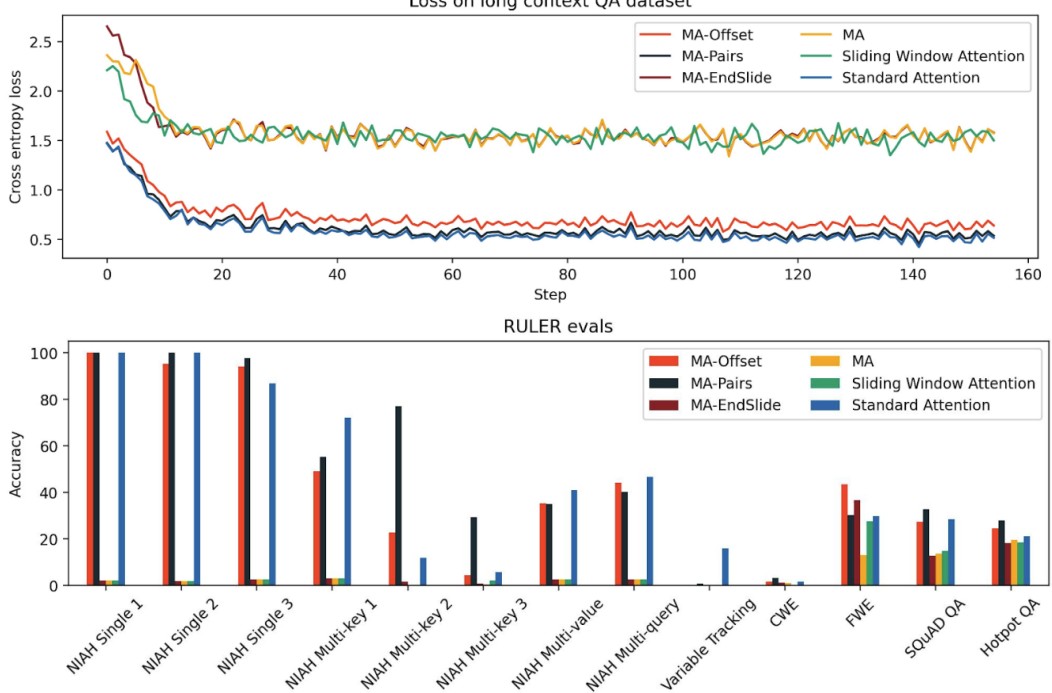

**Figure 4:** Effect of Standard Attention Layers: (Top) Loss curves of the models when fine tuning on long context QA dataset. (Bottom) RULER evals for the models. MA and MA-EndSlide perform poorly on long context tasks whereas MA-Offset and MA-Pairs perform well. This indicates that having a standard attention KV cache which is computed in later layers is important for long context abilities. We also found that the loss on long context QA dataset correlates well with the model's long context abilities.

Both MA and MA-EndSlide have only one standard attention KV-cache, which is computed in the first layer, whereas both MA-Offset and MA-Pairs have at least one standard attention KV-cache which is computed in deeper layers. Hence, this indicates that having at least one standard attention KV cache that is computed in the deeper layers of a transformer model is necessary for good long-context abilities.

## 5.2 KV CACHE SHARING IN SLIDING WINDOW LAYERS

We found that increasing the sharing between sliding window layers (Figure 5) degraded the model's long context performance: MA-Offset-SlideShare was worse than MA-Offset and MA-Pairs-SlideShare was worse than MA-Pairs (Figure 6). This shows that the KV cache sharing pattern amongst the sliding window layers is also important for long context abilities. We have provided some more ablation experiments in the appendix.

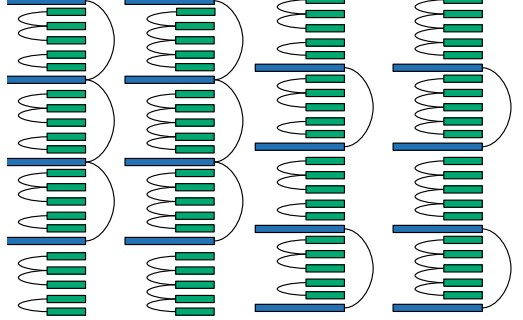

**Figure 5:** Increasing KV cache sharing in sliding window layers: To measure the effect of KV cache sharing in the sliding window layers, we compared the architectures shown in the figure above.

## 5.3 GAUNTLET EVALS

Using the Mosaic Eval Gauntlet v0.3.0 (Mosaic, 2023a), we measured the performance of MixAttention models on standard tasks like MMLU (Hendrycks et al., 2021), HellaSwag (Zellers et al., 2019), etc. to verify that they retain good shorter context abilities. All of the tasks in this eval suite have context lengths of less than a few thousand tokens.

We found that MixAttention models have similar eval metrics to the baseline model on commonsense reasoning, language understanding, and world knowledge. However, we see that they perform

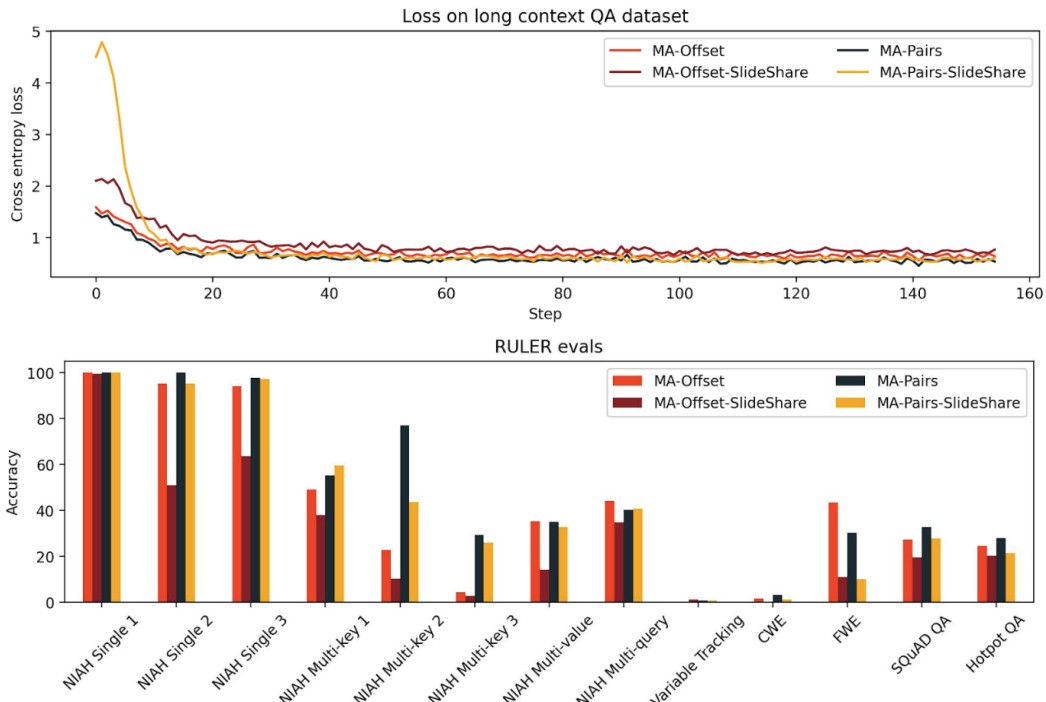

**Figure 6:** Effect of increasing KV cache sharing in sliding window layers: (Top) Loss curves of the models when fine tuning on long context QA dataset. (Bottom) RULER evals for the models. We found that increasing the KV cache sharing in sliding window layers worsened long context abilities of MixAttention Models.

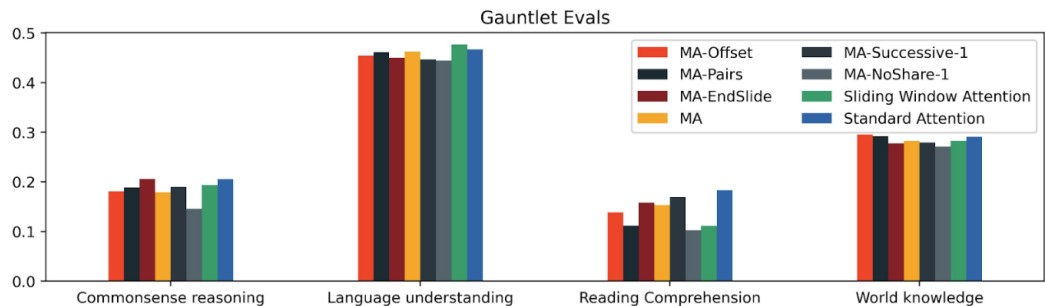

**Figure 7:** Performance of MixAttention models on the Eval Gauntlet: We found that MixAttention models have similar eval metrics to the baseline model on commonsense reasoning, language understanding, and world knowledge. However, we see that they perform worse on reading comprehension.

worse on reading comprehension. An interesting open question is if a different MixAttention configuration or training MixAttention models longer can recover the reading comprehension abilities.

## 5.4 INFERENCE SPEED AND MEMORY CONSUMPTION

We benchmarked the inference speed and memory consumption of MixAttention models by deploying them on a single NVIDIA H100 GPU using SGLang and querying them with 300 prompts, with input length 31000 and output length 1000. In Figure 8, we see that the inference speed of MixAttention models is much faster than standard attention models. We also see in Figure 8 that with MixAttention, we can support a much larger inference batch size in terms of total number of tokens.

Note that the implementation of Sliding Window Attention in SGLang at the time of writing this paper did not optimize the memory consumption for sliding window attention; hence in Figure 8,

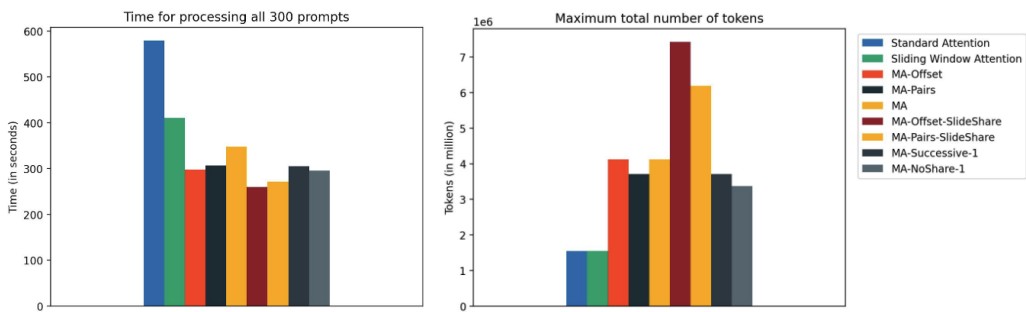

**Figure 8:** Inference with MixAttention: (Left) MixAttention models have significantly faster inference than standard transformers. (Right) MixAttention models can support more tokens, and thus larger batch sizes, during inference.

sliding window attention has the same maximum number of tokens as the standard attention model. Optimizing the memory consumption for sliding window attention should further increase the maximum number of tokens that MixAttention can support during inference.

## 6 CONCLUSION

We find that MixAttention models are competitive with standard attention models on both long- and short-context abilities while being faster during inference and supporting larger batch sizes. We note that on some long context tasks like Variable Tracking and Common Word Extraction, neither MixAttention nor standard attention models perform well. We believe this was because our models weren't trained long enough or the models need a different kind of long context data to be trained for such tasks. More research needs to be done to measure the impact of MixAttention architectures on such metrics.

We encourage others to explore more MixAttention architectures to learn more about them. Below are a few observations to help with further research:

- Adding a standard attention layer in the initial layers by itself does not seem to help long context abilities (for example, see MA-NoShare-1 in the appendix), even if the KV cache from that layer is reused in layers deeper into the network (MA and MA-EndSlide). Hence we recommend placing the first standard attention layer deeper in the network (like MA-Offset) or having multiple standard attention layers, at least one of which is computed at a deeper layer (like MA-Pairs).

- Sliding window layers also contribute to the model's long context abilities. Increasing the KV cache sharing amongst the sliding window layers worsened long context abilities (MA-Offset-SlideShare and MA-Pairs-SlideShare). For that reason, we think that the 2-3 sharing pattern in sliding window layers (Character.AI, 2024) seems to strike a good balance.

- Sharing standard attention KV caches between consecutive layers gave mixed results, with slightly worse accuracy on long context QA tasks (see the appendix).

- In our experiments, MA-Offset and MA-Pair showed great speedup and memory savings during inference, while also maintaining long and short context abilities. Hence, MA-Offset and MA-Pairs might be good configurations for further research.

In general, there is a large hyperparameter space to explore, and we look forward to seeing a variety of new strategies for reducing the cost of inference via combinations of sliding window attention and KV cache reuse.

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
