# OpenReview forum: "Inference-Friendly Models With MixAttention"
_ICLR.cc/2025/Conference — Submitted to ICLR 2025_

### Official Review · Reviewer_mii4 · 2024-10-20

**Soundness:** 3
**Presentation:** 2
**Contribution:** 1
**Rating:** 1
**Confidence:** 5

**Summary:**

The paper proposed an approach called MixAttention which is interleaving standard attention with sliding window attention. Their MixAttention approach also shares KV-cache across the layers. All these optimizations lead to reduce memory usage for the model during inference without significantly deteriorating the model accuracy.

**Strengths:**

The paper is easy to follow and unlike most approaches that use custom device-level code to make inference efficient, the approach doesn't require any custom kernels. This makes the approach easier to adapt to slight changes in the model architecture or running inference on hardware from other vendors.

**Weaknesses:**

1. There is no novelty in the approach. The paper just evaluates the approach proposed in the [blog](https://research.character.ai/optimizing-inference/) by character.AI with slight modifications. Also, there is nothing new written in the paper different from the blog.
2. The authors have not put in enough effort for the paper. There is no optimization done in SGLang to optimize the inference for sliding window attention baseline.
3. The paper is poorly written and there are some typos in the paper. For instance, line 199 uses the word 'sequence' twice in succession.
4. The paper also says to refer to the appendix for a few experiments, however, there is no appendix in the paper.
5. I don't believe that any amount of experiments can make the paper in an acceptable format since there is no novelty.

**Questions:**

1. There is no Pareto improvement shown. How does the proposed approach compare to a smaller standard MoE model with similar KV-cache size? It would be ideal to see a Pareto-improvement curve with KV-cache memory on the X-axis and model accuracy on Y-axis.

---

### Official Review · Reviewer_fhax · 2024-11-03

**Soundness:** 2
**Presentation:** 2
**Contribution:** 2
**Rating:** 3
**Confidence:** 4

**Summary:**

This paper ablates over a particular modification to the transformer architecture where kv-caches are shared across layers and a portion of layers use sliding window attention, for the purpose of reducing compute and memory while retaining performance.
Their main findings show that sharing the KV-cache from the first layer, throughout the entire network hurts performance on RULER (at 32k ctx), and so the KV-cache for a non-sliding window attention layer should be computed at least once in deeper layers, while also controlling for the level of kv-cache sharing on the sliding window attention layers.

**Strengths:**

- Cache sharing across layers has not been extensively studied and ablated over, and so this paper provides additional sample points that show the relationship between cache sharing approach and performance.
- The authors tested their results on RULER which is a long-context benchmark and more conventional evals such as MMLU and HellaSwag through the Gauntlet evals framework which unveils differences in performance between different KV-cache sharing approaches.
- Some of these KV-cache sharing variants perform as well as standard attention while being significantly cheaper in compute and memory.

**Weaknesses:**

- Lack of insight or discussion as to why certain cache-sharing approaches perform better or worse.
- The paper lacks novelty, as it mostly relies on architectural configurations proposed by a blog by CharacterAI [1], and as a consequence, it lacks explanation as to why these configurations were selected in the first place.
- In general, the main critique is that the paper presents only surface level analysis of the observations and does not contribute much to a deeper understanding of why certain cache-sharing approaches perform better than others.

[1] Character.AI. Optimizing AI Inference at Character.AI — research.character.ai. https://research.character.ai/optimizing-inference/, 2024.

**Questions:**

- It would be interesting to see trends between performance and degree of cache-sharing for both standard attention and sliding window attention, as this would give us a better understanding of the rate at which the performance worsens.
- More explanation for why certain choices were made for the experiments such as the eval benchmark of choice, selection of cache-sharing variants.
- More discussion and analysis of the results that leads to deeper insights.
- More discussion about the differences between this and the other cache-sharing paper [1].

[1] William Brandon, Mayank Mishra, Aniruddha Nrusimha, Rameswar Panda, and Jonathan Ragan Kelly. Reducing transformer key-value cache size with cross-layer attention. arXiv preprint arXiv:2405.12981, 2024.

---

### Official Review · Reviewer_wFCF · 2024-11-03

**Soundness:** 2
**Presentation:** 3
**Contribution:** 2
**Rating:** 3
**Confidence:** 5

**Summary:**

This paper aims to optimize the inference efficiency of LLMs by reducing the amount of KV cache. The core intuition of this paper is to combine two existing approaches, i.e., sliding window attention and layer-wise sharing of KV cache, to further reduce the memory cost of inference. Although this kind of combination has already been proposed by some blog and papers, this paper aims to explore the effectiveness of this kind of method from an empirical perspective.

**Strengths:**

1. The combination of sparsifying the token of sequence and sharing the KV cache across layers seems to be a promising method to reduce the inference cost. This paper conducts some interesting experiments, from pre-training to evaluation, to give us some insights regarding the impact of different choices of the setups of such combination.
2. The experiment setup is reasonably designed.

**Weaknesses:**

1. The novelty is limited in two ways. Firstly, it is a straightforward combination of two existing techniques without many adjustments. Secondly, this combination has already been explicitly described in the blog of character.ai, as cited by the authors.
2. I can get that the value of this paper is to provide some empirical guidelines of this combination method, but still, the new information brought by this paper is also limited. For example, “…having the standard KV cache computed in the deeper layers is more important for long context abilities than the standard KV cache of the first few layers.” has been declared by some existing studies. In general, the experiment conclusions of this paper are some high-level phenomenons, instead of some practical methodology.
3. The experiments are all based on a 5B MoE model, which makes the generalisability of the conclusions less convincing.
4. There are quite a few new hyper-parameters getting involved, e.g., for a N-layer model, how to decide which layers are standard attention, which layers are sliding window? how many layers for a KV-sharing group? These decisions are pre-defined in this paper, but what’s really interesting is how to make these decisions wisely given a new model.

**Questions:**

Please refer to the Weakness part

---

### Official Review · Reviewer_xieg · 2024-11-03

**Soundness:** 2
**Presentation:** 1
**Contribution:** 1
**Rating:** 1
**Confidence:** 4

**Summary:**

The authors introduce MixAttention, an architecture that employs sliding window attention to store only recent tokens while sharing KV caches across layers. They train and evaluate four different variants and report the corresponding results.

**Strengths:**

The idea is simple and clear, the experimental setup is also quite clear.

**Weaknesses:**

1. This paper lacks innovation; both the recent window and multi-layer attention are established techniques. The paper simply combines these two methods without any improvements.

2. The experimental results are presented solely as bar charts. I believe it would be beneficial to include a table with some precise values.

3. This paper resembles more of a technical report rather than an innovative and well-developed research paper,  which does not meet the high standards of ICLR.

**Questions:**

Refer to the weaknesses.

---

### Meta-Review · Area_Chair_vm6X · 2024-12-20

**Metareview:**

The paper proposes MixAttention, an architectural modification for language models that combines sliding window attention with KV cache sharing across layers to reduce memory usage and improve inference speed.

The main concerns were the lack of novelty as it primarily reproduced and evaluated ideas already published in a blog post. Additionally, reviewers noted that combining existing techniques (sliding window attention and KV cache sharing) without meaningful improvements or deeper analysis did not yet meet ICLR's standards.
We hope the feedback helps to strengthen the paper for a future occasion.

**Additional Comments On Reviewer Discussion:**

Authors did not provide a response in the feedback phase, so this remained a clear case.

---

### Decision · Program_Chairs · 2025-01-22

Reject